# Magnetotail reconnection onset caused by electron kinetics with a strong external driver

San Lu[1,14], Rongsheng Wang [2,3,14], Quanming Lu [2,3✉], V. Angelopoulos [1], R. Nakamura[4],
A. V. Artemyev[1,5], P. L. Pritchett[6], T. Z. Liu[1], X.-J. Zhang [1], W. Baumjohann [4], W. Gonzalez[7], A. C. Rager[8],
R. B. Torbert [9], B. L. Giles[10], D. J. Gershman[10], C. T. Russell[1], R. J. Strangeway[1], Y. Qi[1], R. E. Ergun[11],
P.-A. Lindqvist [12], J. L. Burch [13] & Shui Wang[2,3]

Magnetotail reconnection plays a crucial role in explosive energy conversion in geospace. Because of the lack of in-situ spacecraft observations, the onset mechanism of magnetotail reconnection, however, has been controversial for decades. The key question is whether magnetotail reconnection is externally driven to occur first on electron scales or spontaneously arising from an unstable configuration on ion scales. Here, we show, using spacecraft observations and particle-in-cell (PIC) simulations, that magnetotail reconnection starts from electron reconnection in the presence of a strong external driver. Our PIC simulations show that this electron reconnection then develops into ion reconnection. These results provide direct evidence for magnetotail reconnection onset caused by electron kinetics with a strong external driver.

[1] Department of Earth, Planetary, and Space Sciences and Institute of Geophysics and Planetary Physics, University of California, Los Angeles, CA, USA. [2] CAS Key Laboratory of Geospace Environment, Department of Geophysics and Planetary Science, University of Science and Technology of China, Hefei, Anhui, China. [3] CAS Center for Excellence in Comparative Planetology, Hefei, Anhui, China. [4] Space Research Institute, Austrian Academy of Sciences, Graz, Austria. [5] Space Research Institute, Russian Academy of Sciences, Moscow, Russia. [6] Department of Physics and Astronomy, University of California, Los Angeles, CA, USA. [7] China-Brazil Joint Laboratory for Space Weather, Instituto Nacional de Pesquisas Espaciais, São Jose dos Campos, SP, Brazil. [8] Department of Physics, Catholic University of America, Washington, DC, USA. [9] University of New Hampshire, Main Campus, Durham, NH, USA. [10] Heliophysics Science Division, NASA Goddard Space Flight Center, Greenbelt, MD, USA. [11] Department of Astrophysical and Planetary Sciences, University of Colorado, Boulder, CO, USA. [12] School of Electrical Engineering, Royal Institute of Technology, Stockholm, Sweden. [13] Southwest Research Institute, San Antonio, TX, USA. [14] These authors contributed equally: San Lu, Rongsheng Wang. ✉email: qmlu@ustc.edu.cn

Magnetic reconnection, a plasma process that converts magnetic energy to particle energy via topological changes in magnetic field lines[1,2], is widely believed to cause explosive phenomena in space and astrophysical plasmas. Magnetic reconnection in Earth's magnetotail plays a crucial role in the most important explosive phenomena in Earth's magnetosphere, geomagnetic storms, and substorms[3,4]. Although magnetotail reconnection has been observed by spacecraft for decades[5–8], how it is even started has been a conundrum because it should be prevented from happening by the normal magnetic field $B_N$ in the magnetotail[9–12].

To justify the occurrence of magnetotail reconnection, for >50 years, theoretical and simulation efforts have been made to explain the onset of magnetotail reconnection from a quiescent current sheet. These efforts have led to two distinct evolutionary mechanisms. In mechanism I, magnetotail reconnection starts from small-scale reconnection caused by electron kinetics[13] (electron reconnection); in mechanism II, ion kinetics initiates magnetotail reconnection directly on a larger scale[14]. Both mechanisms circumvent the stabilizing effect of $B_N$: in mechanism I, a strong external driver reduces $B_N$ to a small enough value so that electron kinetics is allowed to initiate magnetotail reconnection[15–20]; in mechanism II, a hump of $B_N$ renders magnetotail reconnection initiated by ion kinetics[21–24].

Which of the above two mechanisms is responsible for the onset of magnetotail reconnection is still hotly debated. To distinguish between them, it is crucial to determine whether electron reconnection exists in the magnetotail and whether it exists along with a strong external driver. Because electron reconnection is transient and confined to a small region[25–28], previous attempts to detect it in the magnetotail have been limited by the resolution of spacecraft measurements.

Here we report detection of electron reconnection in the magnetotail using high-resolution measurements by the Magnetospheric Multiscale (MMS) spacecraft[29], and further analysis shows that the electron reconnection does occur within a strongly externally driven environment, which provides experimental evidence for the onset of magnetotail reconnection caused by electron kinetics with a strong external driver.

## Results

**Electron reconnection detected by MMS**. The electron reconnection detected by MMS from 20:24:03 to 20:24:11 UT on 17 June 2017 is shown in Fig. 1. The data are presented using a local coordinate system, *LMN* (see "Local coordinate system for the MMS event" in "Methods"). The crossing of the current sheet along the *N* (normal) direction is shown by the reversal of $B_L$ in Fig. 1a. The normal magnetic field $B_N$ is very weak (Fig. 1a), suggesting that it has been locally reduced. Such a reduced $B_N$ favors the onset of electron reconnection via electron Landau resonance in the current sheet. The MMS spacecraft also detects signatures of the Hall magnetic field $B_M$ (Fig. 1a) and the Hall electric field $E_N$ (Fig. 1e). The current sheet crossing velocity, determined by the four-spacecraft timing method[30], is about 67 km/s. The duration of the current sheet crossing lasts about 1.5 s. Therefore, the current sheet's half-width is about 50 km, about $7d_e$, where $d_e = 7.2$ km is the electron inertial length evaluated using the measured plasma density at the center of the current sheet (~0.55 cm$^{-3}$). In this electron-scale current sheet, the very intense cross-tail current density, $j_\perp \approx 200$ nA/m$^2$, is mainly carried by electrons with $V_{eM} \approx 2000$ km/s (Fig. 1c, d). During the current sheet crossing, $B_L$ changes from about $-12$ nT to about 15 nT (Fig. 1a), so the asymptotic magnetic field is $B_0 = 12$–15 nT. Therefore, the Alfvén velocity is about 353–440 km/s, evaluated using $B_0$ and the density of about 0.55 cm$^{-3}$ at the

center of the current sheet. A super-Alfvénic electron outflow $V_{eL} \approx 1000$ km/s is observed at the center of the electron current sheet (Fig. 1c). Ions do not respond to this electron-scale current sheet; the ion flows are weak and remain unchanged during this crossing (Fig. 1b).

Electron acceleration is observed at the center of this electron current sheet (Fig. 1f); no ion acceleration is found across the current sheet (Fig. 1g). As shown in Fig. 1h–j, the discrepancy between the electron flow velocity $\mathbf{V}_e$ and the electric drift velocity $\mathbf{E} \times \mathbf{B}/B^2$ at the current sheet's center indicates that the electrons there are demagnetized. The agyrotropic (or crescent shaped, Fig. 1k) electron velocity distribution at the current sheet center also demonstrates electron demagnetization and kinetic effects[31–33]. The super-Alfvénic electron outflow and the electron acceleration and demagnetization all suggest the occurrence of electron reconnection. This will be further confirmed by the electron heating and the non-zero $\mathbf{j} \cdot \mathbf{E}'$ at the electron reconnection site.

**Strong external driver**. To investigate whether this electron reconnection occurs within a strongly externally driven environment, we use three parameters in Fig. 2 to show the global context of the 4 h around the MMS electron reconnection event. The first parameter is the north–south component of the interplanetary magnetic field (IMF). When the IMF is southward, it reconnects with the dayside northward geomagnetic field, which opens up the magnetosphere, leading to strong global convection and thinning of the magnetotail current sheet. As shown in Fig. 2a, the IMF is (on average) southward for over 2 h prior to and during this electron reconnection event. The second parameter is the auroral electrojet (AE) index used to indicate the intensity of activity in geospace. Figure 2b shows that the AE index increases gradually then abruptly from about 100 to 800 in about 2 h, which is a typical signature of the growth and expansion phases of a magnetospheric substorm[3]. The MMS event occurs in the onset of the substorm's expansion phase, which involves strong global convection and thinning of the magnetotail current sheet[34]. The third parameter is the magnetic field amplitude in the tail lobes—the upstream region of magnetotail reconnection. The lobe magnetic field's magnitude is calculated using MMS data based on the pressure balance, $B_{\text{lobe}} = \left( B_x^2 + B_y^2 + 2\mu_0 p_p \right)^{1/2}$, where the plasma pressure $p_p = n_e T_e + n_i T_i$. Figure 2c shows that the lobe magnetic field peaks just when the electron reconnection event is observed, and its peak value of about 40 nT is even higher than typical substorm time values[35,36], which means that the upstream region of the electron reconnection is strongly compressed due to the strong global convection. Immediately after this event of electron reconnection in the magnetotail, the lobe field decreases because the onset of reconnection dissipates the upstream magnetic field[37,38]. The above analysis shows that the electron reconnection event (20:24:03–20:24:11 UT on 17 June 2017) occurs in the presence of a strong external driver.

**Particle-in-cell (PIC) simulations**. To better understand the above observational results, we use PIC simulations to examine the onset of magnetotail reconnection caused by electron kinetics with a strong external driver, as shown in Fig. 3 (for the simulation set-up, see "PIC simulation model" in "Methods"). Because of the strong driver, the normal magnetic field $B_N$ is reduced to almost zero at $t = 52\Omega_{i0}^{-1}$, and a thin current sheet is formed (Fig. 3a). Reconnection then occurs in this thin current sheet, starting from a mild, small-scale electron phase, at $t = 63\Omega_{i0}^{-1}$. The occurrence of reconnection is justified by the topological

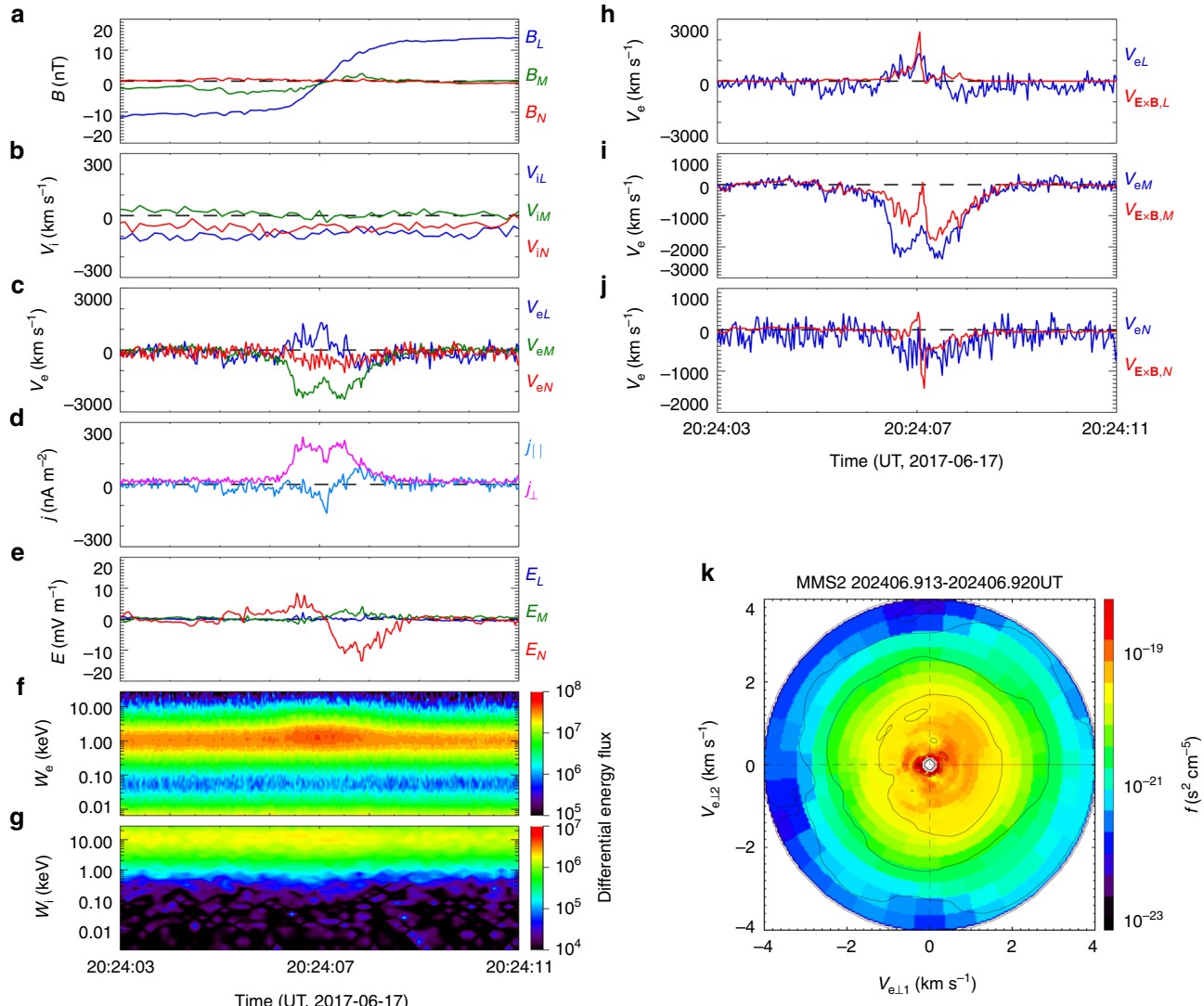

**Fig. 1 MMS2 observations of electron reconnection in Earth's magnetotail.** The event was observed at $[-19.4, -10.4, 5.5]R_E$ from 20:24:03 to 20:24:11 UT on 17 June 2017. **a** Magnetic field components; **b** ion bulk velocity; **c** electron bulk velocity; **d** current densities parallel and perpendicular to the magnetic field (here the current density is calculated using $\mathbf{j} = en_e(\mathbf{V}_i - \mathbf{V}_e)$); **e** electric field; **f, g** electron and ion differential energy flux (in unit of keV cm$^{-2}$ s$^{-1}$ sr$^{-1}$ keV$^{-1}$) as function of electron energy $W_e$ and ion energy $W_i$, respectively; **h–j** electron bulk velocity $\mathbf{V}_e$ and the electric drift velocity $\mathbf{E} \times \mathbf{B}/B^2$; **k** electron velocity distribution in the plane perpendicular to the magneticfield ($\mathbf{v}_{e\perp 1} = [(\mathbf{B} \times \mathbf{v}_e) \times \mathbf{B}]/B^2$, $\mathbf{v}_{e\perp 2} = (\mathbf{B} \times \mathbf{v}_e)/B$) at the electron reconnection site, from 20:24:06.913 to 20:24:06.920 UT. Photo electrons with energies <96 eV, as shown in **f**, are insignificant contaminants, so they are excluded in the calculation of electron bulk velocity.

change in magnetic field lines (Fig. 3b) and the slight increase in the reconnection electric field $E_M$ at the reconnection site $L \approx -16.1d_i$ (Fig. 3f). In the electron phase (compared to the pre-reconnection phase), only electron flows (in the $L$ and $M$ directions) increase (Fig. 3d, g), whereas ion flows do not change (Fig. 3e, h), indicating that ion dynamics are not affected. Reconnection in the electron phase is referred to as electron reconnection, and because ions barely participate in electron reconnection, it is also referred to as electron-only reconnection[25–28]. This electron phase reconnection then develops into a faster, larger-scale ion phase reconnection, e.g., at $t = 90\Omega_{i0}^{-1}$ (Fig. 3c), with a typical fast reconnection rate[39] of $E_M/(V_A B_0) \sim 0.1$ (Fig. 3f). In the ion phase reconnection, the ion dynamics become well developed, as shown by the much faster ion flows (Fig. 3e, h); the electron flows grow even faster in reaction to this rapid reconnection (Fig. 3d, g). Figure 3i shows $\mathbf{j} \cdot \mathbf{E}'$ (where $\mathbf{E}' = \mathbf{E} + \mathbf{V}_e \times \mathbf{B}$), which is around zero before reconnection (black curves) and then increases slightly in the electron phase (red

curves) and significantly in the ion phase (blue curves) at the reconnection site. Therefore, our PIC simulations show that magnetotail reconnection with a strong external driver starts from electron reconnection and then develops into bursty ion reconnection.

Figure 4 shows a side-by-side comparison of the electron reconnection observed by MMS and that obtained from our PIC simulations. In general, the observations and simulations show good agreement. The half-width of the reconnection electron current sheet in the PIC simulations, about $0.25$–$0.5d_i$ (Fig. 4j), i.e., $5$–$10d_e$ ($m_i/m_e = 400$ in the simulations, so $d_i = 20d_e$), is consistent with the value of about $7d_e$ in the MMS event. The simulations reproduce the earthward electron outflow $V_{eL}$, the Hall magnetic field $B_M$, and the Hall current $j_L$ well (Fig. 4l, i, j, respectively). The current density in the $M$ direction, mostly contributed by the electron flow, $V_{eM} \approx 0.05 V_{eA}$ (Fig. 4l), is consistent with the value of $V_{eM} \approx 2000$ km/s $\sim 0.1 V_{eA}$ (Fig. 4d) from the observations ($V_{eA}$ is the electron Alfvén velocity). Ions

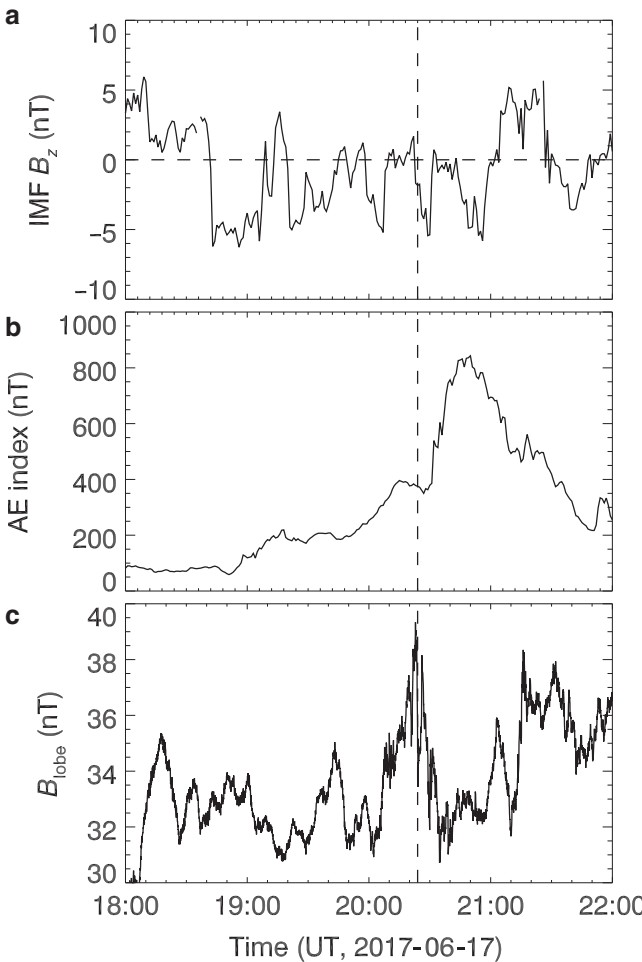

**Fig. 2 Global context of the MMS electron reconnection event. a** OMNI $z$ component of the interplanetary magnetic field (IMF) in the geocentric solar magnetospheric (GSM) coordinate system; **b** auroral electrojet (AE) index; **c** magnitude of magnetic field in the tail lobes. The lobe magnetic field's magnitude is calculated using MMS data based on the pressure balance, $B_{lobe} = \left( B_x^2 + B_y^2 + 2\mu_0 p_p \right)^{1/2}$, where the plasma pressure $p_p = n_e T_e + n_i T_i$. The vertical dashed line represents the time (20:24 UT) when the MMS event of electron reconnection is observed.

barely react to the electron reconnection, as shown by the weak ion flows (Fig. 4c, k) and the constant ion temperature across the reconnection region (Fig. 4f, n). The electrons are heated, especially in the parallel direction (Fig. 4g, o), indicating that electrons react to the electron reconnection. Note that, in the MMS observations, the current density $j_M$ is bifurcated (Fig. 4b) because the electron flow velocity $V_{eM}$ is bifurcated (Fig. 4d). The $V_{eM}$ bifurcation could be caused by the electrons' electric drift and the drift due to the electron parallel temperature anisotropy[40] ($T_{e\parallel} > T_{e\perp}$, see Fig. 4g). The current sheet bifurcation also presents in the simulations but is less pronounced (Fig. 4j, l), possibly because the electron parallel temperature anisotropy is less pronounced in the simulations (Fig. 4o). In both observations and simulations, $\mathbf{j} \cdot \mathbf{E}'$ is nonzero at the reconnection site (Fig. 4h, p), which confirms the crossing of electron reconnection region.

## Discussion

The magnetotail current sheet is usually in the equatorial ($x$–$y$) plane, and its normal is in the $z$ direction in the geocentric solar ecliptic (GSE) coordinate systems. The current sheet observed in

the MMS event, however, is unusual—it is almost in the $x$–$z$ plane (its normal $N = (-0.3076, 0.9506, -0.0408)$ is almost in the $y$ direction in the GSE coordinate system, see "Local coordinate system for the MMS event" in "Methods"). Such a tilted current sheet has been observed in the magnetotail and is believed to be caused by the magnetotail dynamic flapping motion[41,42]. The tilted current caused by flapping motion, however, cannot be resolved by our two-dimensional PIC simulations, therefore, in this study, we use a local $LMN$ coordinate system and focus on local dynamics of this current sheet. Note that, even with the help of three-dimensional simulations, what causes the flapping current sheet is still an open question[43,44], not to mention how the flapping current sheet affects the process of magnetotail reconnection—they both require further investigation.

The MMS electron reconnection event in the magnetotail was reported previously as an electron-scale current sheet without bursty reconnection signatures[45]. Further analysis and comparison with PIC simulations allow us to better understand this event: it is the beginning of magnetotail reconnection when reconnection was in an early stage on a small, electron scale. Ions are too heavy to react to this early-stage reconnection; only electrons react to this electron reconnection. According to our simulations, this electron reconnection would grow into an ion reconnection, which is bursty and on a larger, ion scale (see Fig. 3). Although the MMS spacecraft did not directly encounter a bursty ion reconnection following the electron reconnection, the decrease in the lobe magnetic field (see Fig. 2c) indicates that there was a bursty ion reconnection dissipating the magnetic field around the spacecraft[37,38]. At the center of the bursty ion reconnection region, there is an electron diffusion region (EDR)[5,46], and the EDR has already been observed by the MMS spacecraft in the magnetotail[8]. However, the EDR (in ref. [8]) is different from the electron reconnection (in the present study): the EDR is embedded in the bursty ion reconnection region, so both electron and ion dynamics can be observed; the electron reconnection occurs before ion reconnection emerges, so only electron dynamics can be observed.

The onset of electron reconnection is through an electron-tearing mode instability caused by electron kinetics[13] when $B_N$, which stabilizes the electron-tearing mode[10], is reduced by a strong external driver[15–20]. The onset criterion is given in ref. [19] as

$$\frac{B_N}{B_0} \frac{\delta}{d_i} < \frac{f}{2} \sqrt{\frac{m_e T_e}{m_i T_i}}, \qquad (1)$$

where $f = k_L \delta$ is about 0.5 for the fastest growing mode[47,48] ($k_L$ is the wave number in the $L$ direction and $\delta$ is the current sheet half-width). In the MMS event, the asymptotic magnetic field magnitude is $B_0 = 12$–15 nT, the current sheet half-width is $\delta \approx 7d_e$, the electron temperature is $T_e \approx 0.8$ keV, and the ion temperature is $T_i \approx 5$ keV at the center of the current sheet, so Eq. (1) gives an instability threshold for the normal magnetic field $B_N$ of 0.171–0.214 nT. At the center of the current sheet ($|B_L| < 3$ nT), the average normal magnetic field is $B_N = 0.148$ nT, smaller than the threshold, which theoretically supports the occurrence of electron reconnection through the electron-tearing mode instability. Strictly speaking, this criterion should be used to examine the pre-reconnection current sheet (corresponding to Fig. 3a), whereas, in the MMS event, electron reconnection has already started in the current sheet (corresponding to Fig. 3b). But the electron reconnection is in a very early stage, and such mild, small-scale electron reconnection barely changes the current sheet parameters, therefore the above feasibility analysis of the electron tearing mode is still valid.

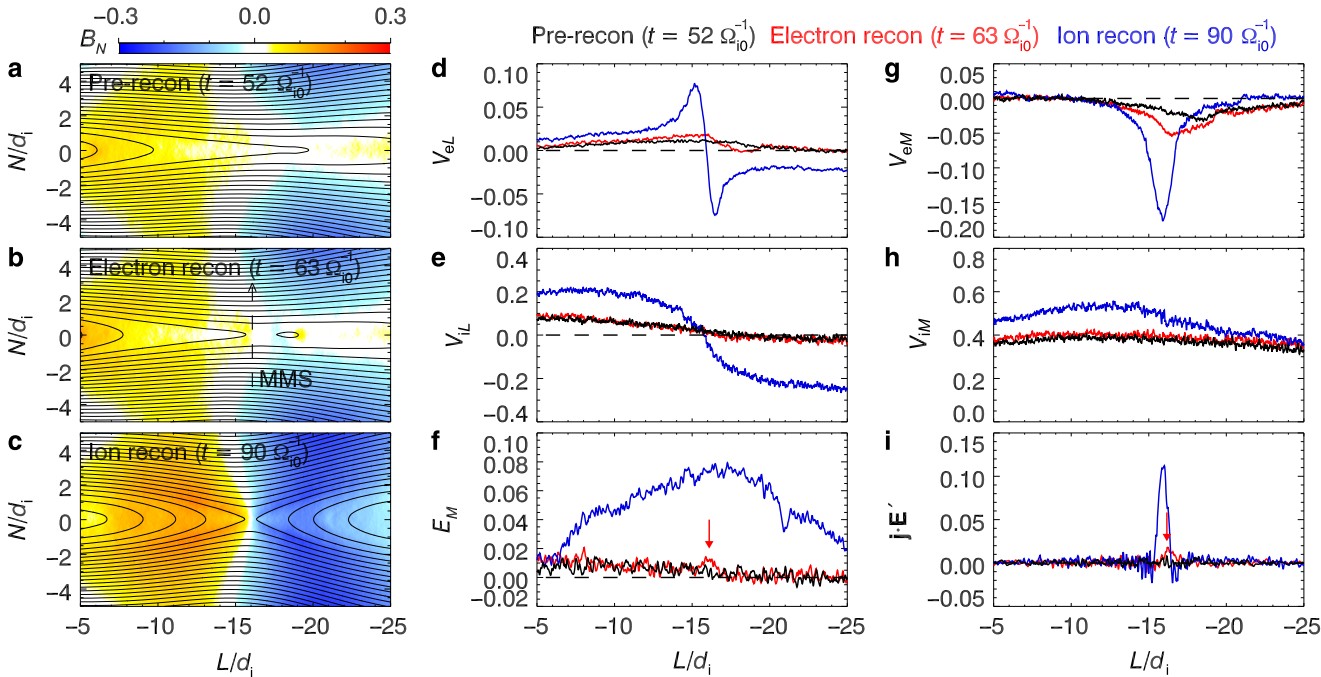

**Fig. 3 PIC simulations of strongly externally driven onset of magnetotail reconnection. a–c** Colors show the normal magnetic field $B_N$ (in unit of $B_0$) in the L–N plane at pre-reconnection phase ($t = 52\Omega_{i0}^{-1}$), electron reconnection phase ($t = 63\Omega_{i0}^{-1}$), and ion reconnection phase ($t = 90\Omega_{i0}^{-1}$), respectively. The black curves represent the magnetic field lines in the reconnection plane. **d–i** Profiles, along $N = 0$, of $V_{eL}$ (in unit of $V_{eA}$), $V_{iL}$ (in unit of $V_A$), $E_M$ (in unit of $V_A B_0$), $V_{eL}$ (in unit of $V_{eA}$), $V_{iL}$ (in unit of $V_A$), and $\mathbf{j} \cdot \mathbf{E'}$ (in unit of $en_0 V_A^2 B_0$) at pre-reconnection phase ($t = 52\Omega_{i0}^{-1}$, black curves), electron reconnection phase ($t = 63\Omega_{i0}^{-1}$, red curves), and ion reconnection phase ($t = 90\Omega_{i0}^{-1}$, blue curves). The dashed line with an arrow in **b** represents the virtual trajectory of the MMS spacecraft across the electron reconnection region, along $L = -16.1d_i$ at $t = 63\Omega_{i0}^{-1}$. The red arrows in **f, i** mark the location of the electron reconnection site.

In addition to the above electron-tearing mode instability, an ion-tearing mode instability has also been proposed to explain the onset of magnetotail reconnection[14]. Although the normal magnetic field $B_N$ can stabilize the ion-tearing mode[11,12], energy principle calculations suggest that this mode can still be unstable when $B_N$ has a hump along the magnetotail[21]. Starting from an initial $B_N$ hump configuration, PIC simulations have been performed using both open outflow[22–24] and closed[49] boundary conditions in the Earth-tail direction. Although there are significant differences regarding the timescales of the resulting instability and formation of subsequent nulls in the $B_N$ field, both simulations find that a characteristic feature of the instability is generation of a strong earthward ion flow with a speed of at least 0.1–0.2 of the ion thermal velocity or Alfvén velocity. This onset mechanism of magnetotail reconnection from a $B_N$ hump configuration, however, is not supported by our MMS event because it lacks such a fast earthward ion flow signature. In our event, the ion flow is weak, constant, and tailward ($V_{iL} < \sim0$, see Fig. 1b). We have already demonstrated that the MMS event does not resemble the EDR in bursty ion reconnection because of the absence of ion dynamics. Therefore, the existence of the non-EDR electron-scale current sheet in the MMS event (half-width $\sim7d_e$) contradicts the ion-tearing mode mechanism because the current sheet would have already been unstable on the ion scale before it thins to the electron scale.

The spatial scale of the electron (-only) reconnection is very small, about seven electron inertial lengths in the MMS event. Moreover, the electron reconnection is transient with a short duration and inconspicuous signatures that could easily be overlooked. Therefore, it is very difficult for spacecraft to encounter the electron reconnection region. Using MMS high-resolution measurements, we report in situ detection of such an electron reconnection in the magnetotail. Further examination

shows that this electron reconnection occurs in conjunction with a strong external driver that reduces the stabilizing $B_N$ to allow reconnection on the electron scale. This study provides experimental evidence for magnetotail reconnection initiated by electron kinetics with a strong external driver, which helps solve the long-standing conundrum of magnetotail reconnection onset.

## Methods

**Local coordinate system for the MMS event.** The MMS2 data are presented in the local (LMN) coordinate system with $L = (0.9477, 0.3023, -0.1029)$, $M = (-0.0855, -0.0703, -0.9939)$, and $N = (-0.3076, 0.9506, -0.0408)$ in the GSE coordinates. The LMN coordinate system is obtained from the minimum variance analysis[50] of the MMS2 magnetic field data from 20:23:30 to 20:24:30 UT on 17 June 2017. The LMN coordinate system is consistent with the one used in the simulations: the reconnection current sheet normal direction is along N, L is the direction of reconnecting anti-parallel magnetic fields, and M is the direction of the current, pointing out of the reconnection plane.

**PIC simulation model.** We use PIC simulations to examine onset of reconnection with an external driver. To be consistent with the MMS observations, an orthogonal coordinate system (L, M, N) is used in the simulations. The simulations are two dimensional, in the L–N plane, and M is the out-of-plane direction. The initial condition is the Lembège–Pellat current sheet[11], which can be described using magnetic vector potential $A_{0M}(L, N) = -B_0\delta\ln\{\cosh[F(L)(N/\delta)]/F(L)\}$ and density $n(L, N) = n_0 F^2(L)\mathrm{sech}^2[F(L)(N/\delta)] + n_b$, where $F(L) = \exp(\epsilon L/\delta)$. Here $\epsilon = B_N/B_0$ represents the constant fraction of the normal magnetic field at $N = 0$, and we use $\epsilon = 0.04$. The current sheet half-width is $\delta = 2d_i$, and $n_b = 0.2n_0$ is the uniform background density. The ion-to-electron mass ratio is $m_i/m_e = 400$, so the ratio between ion and electron inertial lengths is $d_i/d_e = 20$. Uniform initial electron and ion temperatures are adopted, with $T_{i0} = 0.4167m_i V_A^2$ and $T_{e0} = 0.0833m_i V_A^2$, where $V_A$ is the Alfvén velocity evaluated using $B_0$ and $n_0$. The corresponding electron Alfvén velocity $V_{eA} = 20 V_A$. The size of the simulation domain is $-32d_i \le L \le 0$, $-8d_i \le N \le 8d_i$. An external driver is imposed by adding an out-of-plane electric field $E_M = \hat{E}_M(t)\mathrm{sech}^2[(L + 16d_i)/D_L]$ to the upper and lower boundaries at $N = \pm8d_i$, where $\hat{E}_M(t) = 2a\omega B_0 \tanh(\omega t)/\cosh^2(\omega t)$. The parameter $a$ determines the size of the field line deformation region from the upper and lower N boundaries, the parameter $\omega$ dictates the timescale of the driving inflow, and the parameter $D_L$ is the half-width of the driving perturbation in the L direction. We use

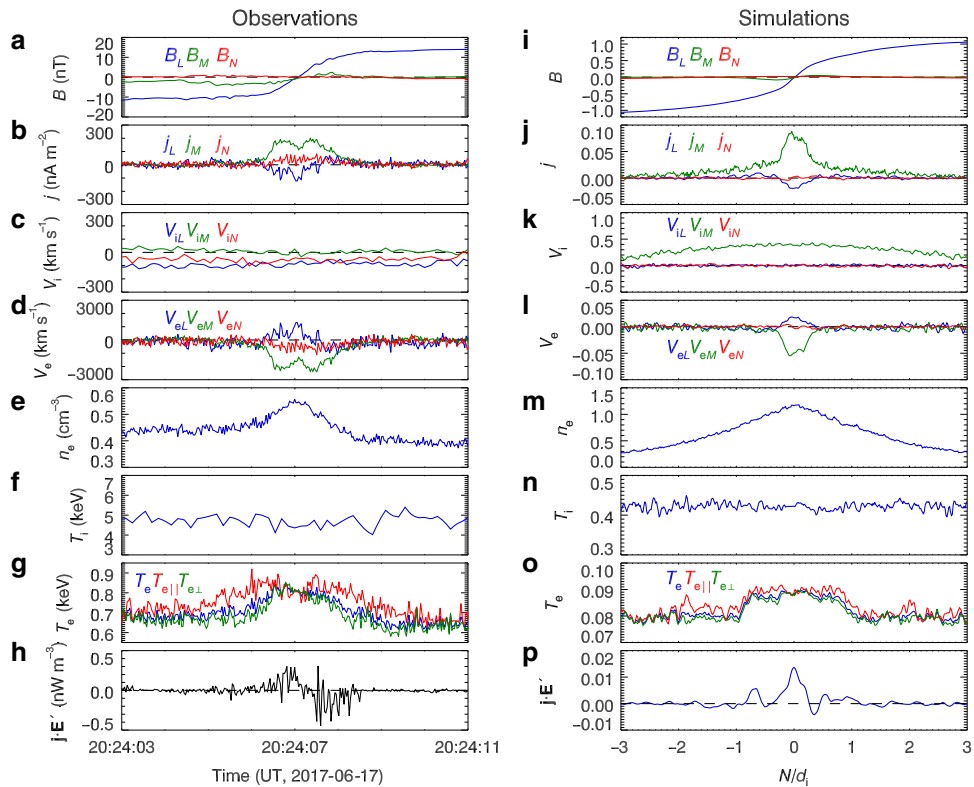

**Fig. 4 Comparison between MMS2 observations and PIC simulations of electron reconnection.** The MMS observations are on the left, and the PIC simulation results along the virtual MMS spacecraft trajectory ($L = -16.1d_i$ at $t = 63\Omega_{i0}^{-1}$, as shown in Fig. 3b) are on the right. **a**, **i** Magnetic field; **b**, **j** current density calculated using $\mathbf{j} = en_e(\mathbf{V}_i - \mathbf{V}_e)$; **c**, **k** ion bulk velocity; **d**, **l** electron bulk velocity; **e**, **m** electron density; **f**, **n** ion temperature; **g**, **o** electron temperature; **h**, **p** energy conversion $\mathbf{j} \cdot \mathbf{E}'$. Photo electrons with energies <96 eV are excluded in the calculation of electron density, bulk velocity, and temperature. In the PIC simulations, magnetic field is in unit of $B_0$, current density is in unit of $en_0V_{eA}$, ion bulk velocity is in unit of $V_A$, electron bulk velocity is in unit of $V_{eA}$, electron density is in unit of $n_0$, electron and ion temperatures are in unit of $m_iV_A^2$, and $\mathbf{j} \cdot \mathbf{E}'$ is in unit of $en_0V_A^2B_0$.

$a = 2d_i$, $D_L = 5d_i$, and $\omega = 0.05\,\Omega_{i0}$, where the ion cyclotron frequency $\Omega_{i0} = eB_0/m_i$. These parameters give a strong driver that enables the externally driven reconnection scenario based on the simulations in ref. [19]. Open boundary conditions are used in the $L$ boundaries. The grid size is $\Delta L = \Delta N = d_i/64$, and the number of grids is $N_L \times N_N = 2048 \times 1024$. The speed of light is $c = 40\,V_A$. The time step is $\Delta t = 0.00025\Omega_{i0}^{-1}$. About $3.5 \times 10^8$ particles per species are employed at the initial time.

## Data availability

MMS data are publicly available from http://lasp.colorado.edu/mms/sdc/public/. AE index data are publicly available from http://wdc.kugi.kyoto-u.ac.jp/ae_provisional/. OMNI data are publicly available from CDAWeb (https://cdaweb.gsfc.nasa.gov/index.html/). All data produced by our PIC simulations for this study are available from https://doi.org/10.6084/m9.figshare.12808541.v1.

## Code availability

MMS data, AE index data, and OMNI data have been imported and analyzed using corresponding plug-ins to the SPEDAS analysis platform (http://spedas.org). The computer code of our PIC simulations for this study is available upon request.

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

## Acknowledgements

This work was supported by Key Research Program of Frontier Sciences, CAS (QYZDJ-SSW-DQC010); National Science Foundation of China (NSFC) grants (41674143, 41922030, 41527804, and 41421063); NASA contract NAS5-02099 and NASA grant 80NSSC18K1122; and Austrian Science Fund (FWF) I2016-N20. We thank the NASA High-End Computing (HEC) Program for providing the computer resources.

## Author contributions

S.L. and R.W. performed simulation, spacecraft data analysis, and writing; Q.L. conceived the idea and oversaw the project; V.A., R.N., A.V. A., P.L.P., T.Z.L., X.-J.Z., W.B., W.G., J.L.B., and S.W. contributed to interpretation of simulation and observation results; A.C.R., R.B.T., B.L.G., and D.J.G., plasma data; C.T.R., R.J.S., and Y.Q., magnetic field data; R.E.E., and P.-A.L., electric field data.

## Competing interests

The authors declare no competing interests.
