## [Peer Review File · Nature Communications]

REVIEWER COMMENTS

Reviewer #1 (Remarks to the Author):

Here is the review report for the manuscript "Magnetotail reconnection onset by electron kinetics with a strong external driver," authored by S. Lu et al. This manuscript reports an EDR crossing event observed by MMS on June 17, 2017, where electron-jet is observed in electron kinetic scale while ions appear to be irresponsive to these fine-structures. This manuscript presents an interesting case of electron-only reconnection, as a counterpart of the electron-only reconnection reported by Phan et al. 2018 at magnetosheath downstream of the bow shock at dayside. The authors also find that this electron-only event occurs under a strong solar wind driving, similar to the result from the onset study in previous PIC simulations. The comparison of various quantities between MMS observation and PIC simulation in Fig. 4 of this manuscript is remarkable.

The authors go on to argue that this electron-only reconnection is a transient phase right after the onset of tail reconnection, representing the direct evidence of reconnection onset mechanism by external solar wind driving, that compresses the current sheet until electron tearing becomes unstable. They argue that this event is in less favor to the competing onset mechanism by ion-tearing instability. Distinguishing these two competing theories (electron tearing vs. ion tearing) in observation is an important issue of substorm study. This manuscript is well-written, the event is clearly presented, and the science question is important. I would suggest for publication if the authors can address the following questions.

1. From the comparison in Fig. 4, I judge that the authors think MMS spacecrafts across the diffusion region in the N direction. This means that MMS will not be able to sample the tailward ion flow (caused by ion tearing, if there is any) at downstream. The concern is that ions are expected to be irresponsive inside the diffusion region in general, regardless of the onset mechanism. Some additional analyses or evidence could potentially make the authors' statement on onset much more convincing. While it could be extra work, I suggest the authors conduct a similar simulation using the setup of Sitnov et al. 2011 and examine whether the ion flow V_{iL} does look different inside the diffusion region in such setup. An alternative way to address this question is to properly discuss the ion flow signature inside the diffusion region in Sitnov's or his collaborators' work [for instance, Sitnov et al. (2013), GRL 40, 22]. This will render readers to better assess the difference of ion signature inside the diffusion region from these two competing theories.

2. Line 70 and 131 provides the estimation of the current sheet half-width of $7d_e$. Is it consistent with the onset criterion in Liu et al., 2014 based on electron tearing instability?

3. In line 75, how is the asymptotic magnetic field magnitude 15 nT estimated? Being relevant to this question, then in Fig. 2, how is the lobe field magnitude 40 nT at the same time determined? Both values are derived from MMS observation, but it is not clear how the authors get these two different values.

4. From reading the fluctuation of B_{lobe} in Fig. 2c, should we also expect reconnection in the second peak at 21:15?

Reviewer #2 (Remarks to the Author):

In this paper the authors present evidence from the MMS spacecraft and particle-in-cell simulation data to argue that electron tearing can occur in the magnetotail current sheet in response to strong upstream driving. The electron tearing instability is a leading candidate to explain substorm onset, which is one of the key unsolved problems in space physics. The topic is not only of academic interest, but also of practical interest for space weather modeling. For electron tearing to occur in the tail, the current layer must be sufficiently thin, and the stabilizing influence of the normal magnetic field must be removed.

To fully address this problem is challenging, as it requires both local in-situ data from the diffusion region, as well as a more global picture of the configuration and driving of the magnetotail current sheet. It also requires information across multiple timescales that shows the different phases of reconnection. Overall, I think the evidence the authors present is compelling, although not totally conclusive to confirm this picture of reconnection onset.

In particular, the MMS data shows quite a clear picture of electron-scale reconnection occurring without coupling to ions. It is noted that much of this data has been previously published elsewhere with slightly different interpretation. The data showing prior southward IMF from OMNI and the estimation of the lobe field indicate strong driving, and the Auroral Electrojet index indicates strong geomagnetic activity around the time that this reconnection occurs. However, there is unfortunately no confirmation from MMS that ion-scale reconnection occurs at the same spatial location later on. The authors use 2D kinetic simulations to interpret the electron-scale reconnection phase and argue that it proceeds the (unobserved) later ion-scale phase. The simulated spacecraft traces are in reasonably good agreement with the real data, although there are some noticeable differences – namely a bifurcated current layer with electron anisotropy, which is not seen in the simulations. The authors should comment on possible reasons for this difference.

A more significant issue, which I only realized from reading the previous paper, is that the current layer is quite dynamic, and is actually oriented at almost 90 degrees to the current sheet in the simulation. This is suggestive of some other instability present in the current sheet, which could conceivably play a role in the current sheet thinning process. It is not possible to capture such physics in a 2D simulation and I think it would be unfair to ask for an expensive 3D simulation to be performed, but the authors should at least discuss this important difference as it may affect the conclusions.

A few more minor points:

1. The “Newton Challenge” used a standard Harris sheet current sheet set-up.
2. In the last paragraph of the discussion ‘about several inertial lengths’ should be ‘about seven inertial lengths’.
3. I think the authors should estimate whether the normal field B_N is small enough in the MMS data to remove the electron tearing stabilization mechanism.

Overall, I think the paper is of interest and could be suitable for publication in this journal. The subject problem is of high interest for magnetic reconnection and the wider space physics community. It is particularly challenging to address this problem, but the authors present what is probably the most compelling evidence yet for the driven electron tearing scenario. However, I would like the authors to comment on the differences between the MMS data and their simulations, rather than ignore the limitations. The paper is well written, and contains appropriate references.

Response to Reviewer [NCOMMS-20-13389]

We thank the Reviewers for carefully reading our manuscript and providing helpful comments. The manuscript has been revised to address these comments. The changes are highlighted in red in the manuscript. Our point-by-point responses are given below, and the corresponding revisions are also described.

Reviewer #1:

Here is the review report for the manuscript “Magnetotail reconnection onset by electron kinetics with a strong external driver,” authored by S. Lu et al. This manuscript reports an EDR crossing event observed by MMS on June 17, 2017, where electron-jet is observed in electron kinetic scale while ions appear to be irresponsive to these fine-structures. This manuscript presents an interesting case of electron-only reconnection, as a counterpart of the electron-only reconnection reported by Phan et al. 2018 at magnetosheath downstream of the bow shock at dayside. The authors also find that this electron-only event occurs under a strong solar wind driving, similar to the result from the onset study in previous PIC simulations. The comparison of various quantities between MMS observation and PIC simulation in Fig. 4 of this manuscript is remarkable.

The authors go on to argue that this electron-only reconnection is a transient phase right after the onset of tail reconnection, representing the direct evidence of reconnection onset mechanism by external solar wind driving, that compresses the current sheet until electron tearing becomes unstable. They argue that this event is in less favor to the competing onset mechanism by ion-tearing instability. Distinguishing these two competing theories (electron tearing vs. ion tearing) in observation is an important issue of substorm study. This manuscript is well-written, the event is clearly presented, and the science question is important. I would suggest for publication if the authors can address the following questions.

1. From the comparison in Fig. 4, I judge that the authors think MMS spacecraft across the diffusion region in the N direction. This means that MMS will not be able to sample the tailward ion flow (caused by ion tearing, if there is any) at downstream. The concern is that ions are

expected to be irresponsive inside the diffusion region in general, regardless of the onset mechanism. Some additional analyses or evidence could potentially make the authors' statement on onset much more convincing. While it could be extra work, I suggest the authors conduct a similar simulation using the setup of Sitnov et al. 2011 and examine whether the ion flow ViL does look different inside the diffusion region in such setup. An alternative way to address this question is to properly discuss the ion flow signature inside the diffusion region in Sitnov's or his collaborators' work [for instance, Sitnov et al. (2013), GRL 40, 22]. This will render readers to better assess the difference of ion signature inside the diffusion region from these two competing theories.

Reply: We thank the Reviewer for the helpful suggestion! It is important to discuss why the MMS event supports the electron tearing mechanism but not the ion tearing mechanism. Because reconnection onset through the ion tearing mode instability in the presence of a Bz hump has been well documented by Sitnov's and his collaborators' work, we do not repeat their simulations in the present study. Instead, as suggested by the Reviewer, we have now properly discussed Sitnov et al.'s mechanism and explained why it is not supported by the MMS event. In the discussion, we first describe the mechanism. In Sitnov et al.'s simulations, a characteristic feature of is the generation of a strong earthward ion flow front. Fig. R1 adopted from Fig. 4 in Sitnov et al. (2013) well presents their simulation results. In the simulations, magnetic reconnection occurs at about $x = -12$ according to the signatures of electron flow and normal magnetic field (Figs. R1a, b), and Fig. R1c shows the strong earthward ion flow, which is at least 0.1-0.2 Alfvén velocity at $x = -12$ (note that the negative V_{ix} represents earthward ion flow because the x coordinate in the simulations is opposite to the GSE coordinate). However, in the MMS event, the ion flow is weak, constant, and tailward (see Fig. 1b). Therefore, the MMS event does not support Sitnov et al.'s mechanism. We further point out that the existence of the non-EDR electron current in the MMS event also contradicts the ion tearing mode mechanism because the current sheet would have already been unstable on the ion-scale before it thins to the electron-scale. The above discussion on this issue has now been added into the revised manuscript in lines 199-215.

Fig. R1. Adopted from Fig. 4 in Sitnov et al. (2013). Profiles along $z = 0$ of (a) normal magnetic field B_z , (b) electron flow velocity V_{ex} , and (c) ion flow velocity V_{ix} at representative times.

2. Line 70 and 131 provides the estimation of the current sheet half-width of $7d_e$. Is it consistent with the onset criterion in Liu et al., 2014 based on electron tearing instability?

Reply: We thank the Reviewer for the question! It leads to an important discussion. Yes, the current sheet half-width and other parameters (B_n , B_0 , and electron and ion temperatures) satisfy the onset criterion of the electron tearing mode instability in Liu et al. (2014). This examination shows that the observed current sheet by the MMS spacecraft is unstable to the electron tearing mode instability, which theoretically supports the occurrence of electron reconnection in this current

sheet. A more detailed discussion on this issue has now been added into the revised manuscript in lines 182-198.

3. In line 75, how is the asymptotic magnetic field magnitude 15 nT estimated? Being relevant to this question, then in Fig. 2, how is the lobe field magnitude 40 nT at the same time determined? Both values are derived from MMS observation, but it is not clear how the authors get these two different values.

Reply: We thank the Reviewer for pointing these out! The asymptotic magnetic field 15 nT was read from the profile of B_L . Upon a closer examination, we find during the current sheet crossing, B_L changes from -12 nT to about 15 nT, so we change the asymptotic magnetic field to $B_0 = 12 - 15$ nT to be more accurate. The Alfvén velocity is also changed accordingly. The above changes has been made in the revised manuscript in lines 75-78.

Regarding the calculation of the lobe magnetic field, we use a well-accepted routine based on pressure balance, $B_{lobe} = (B_x^2 + B_y^2 + 2\mu_0 p_p)^{1/2}$, where the plasma pressure $p_p = n_e T_e + n_i T_i$. Note that B_{lobe} is usually larger than the asymptotic magnetic field B_0 because B_0 refers to the magnetic field at the current sheet boundary layer. The calculation of the lobe magnetic field has been described in the revised manuscript in lines 104-106.

4. From reading the fluctuation of B_{lobe} in Fig. 2c, should we also expect reconnection in the second peak at 21:15?

Reply: This is an interesting question! A peak of B_{lobe} indicates that the magnetotail is strongly driven, which favors occurrence of magnetic reconnection. However, to determine whether reconnection really occurs, one needs to examine other measurements around the B_{lobe} peak. We take a quick look at the MMS measurements, as shown in Fig. R2. Marked by the black boxes, we find that there are indeed a fast ion flow V_x (6th panel) and a strong electric field E_y (8th panel) around the B_{lobe} peak at 21:15 UT. These signatures suggest that magnetic reconnection may occur around the second peak at 21:15 UT. Because further analyses of these measurements are out of the scope of the present paper (the present paper focuses on the event at 20:24 UT), we leave them for future studies.

Fig. R2. A “Quicklook” summary plot of the MMS measurements from 20:00 to 22:00 UT on 17 June 2017 downloaded from MMS’s website <https://lasp.colorado.edu/mms/sdc/public/>.

Reviewer #2:

In this paper the authors present evidence from the MMS spacecraft and particle-in-cell simulation data to argue that electron tearing can occur in the magnetotail current sheet in response to strong upstream driving. The electron tearing instability is a leading candidate to explain substorm onset, which is one of the key unsolved problems in space physics. The topic is not only of academic interest, but also of practical interest for space weather modeling. For electron tearing to occur in the tail, the current layer must be sufficiently thin, and the stabilizing influence of the normal magnetic field must be removed.

To fully address this problem is challenging, as it requires both local in-situ data from the diffusion region, as well as a more global picture of the configuration and driving of the magnetotail current sheet. It also requires information across multiple timescales that shows the different phases of reconnection. Overall, I think the evidence the authors present is compelling, although not totally conclusive to confirm this picture of reconnection onset.

In particular, the MMS data shows quite a clear picture of electron-scale reconnection occurring without coupling to ions. It is noted that much of this data has been previously published elsewhere with slightly different interpretation. The data showing prior southward IMF from OMNI and the estimation of the lobe field indicate strong driving, and the Auroral Electrojet index indicates strong geomagnetic activity around the time that this reconnection occurs. However, there is unfortunately no confirmation from MMS that ion-scale reconnection occurs at the same spatial location later on. The authors use 2D kinetic simulations to interpret the electron-scale reconnection phase and argue that it proceeds the (unobserved) later ion-scale phase. The simulated spacecraft traces are in reasonably good agreement with the real data, although there are some noticeable differences – namely a bifurcated current layer with electron anisotropy, which is not seen in the simulations. The authors should comment on possible reasons for this difference.

Reply: We thank the Reviewer for pointing out the issue of bifurcation! Yes, it is clear that in the MMS event, the current density j_M is bifurcated (Figs. 4b) because the electron flow velocity V_{eM} is bifurcated (Fig. 4d). The bifurcation can be caused by the electron temperature

anisotropy, $T_{e\parallel} > T_{e\perp}$ (Fig. 4g), through $V_{eM} = -(\partial B_L / \partial N)(T_{e\parallel} - T_{e\perp}) / (eB^2)$. Note that the current sheet bifurcation also presents in the simulations but is less pronounced (Figs. 4j, l). The profiles of j_M and V_{eM} also depend on the choice of the virtual spacecraft trajectory. In Fig 4, we choose $L = -16.1d_i$ at $t = 63\Omega_{i0}^{-1}$. Here we further show the profiles of j_M and V_{eM} along $L = -15.9d_i$ at $t = 61, 62,$ and $63\Omega_{i0}^{-1}$ in Fig. R3 to show that the bifurcation exists in the simulations (although less pronounced than in the MMS event). The reason for the less pronounced bifurcation in the simulation may be that the electron temperature anisotropy is weaker in the simulations (Fig. 4o). The current sheet bifurcation issue has now been described and discussed in the revised manuscript in lines 147-153.

Fig R3. Profiles of j_M and V_{eM} along $L = -15.9d_i$ at $t = 61, 62,$ and $63\Omega_{i0}^{-1}$

A more significant issue, which I only realized from reading the previous paper, is that the current layer is quite dynamic, and is actually oriented at almost 90 degrees to the current sheet in the simulation. This is suggestive of some other instability present in the current sheet, which could conceivably play a role in the current sheet thinning process. It is not possible to capture such

physics in a 2D simulation and I think it would be unfair to ask for an expensive 3D simulation to be performed, but the authors should at least discuss this important difference as it may affect the conclusions.

Reply: We thank the Reviewer for bringing out this important issue! Yes, the current sheet in the MMS event has an unusual configuration – it is tilted for about 90 degrees. The tilted current sheet can be formed by the dynamic flapping motion of the magnetotail. The Reviewer is correct that our 2-D PIC simulations cannot resolve this flapping current sheet, therefore, we simply use a local *LMN* coordinate system to study its small-scale dynamics. However, even with the help of 3-D simulations, what causes the flapping current sheet is still an open question, not to mention how it affects the process of magnetic reconnection. As the Reviewer suggested, we have now discussed the issue of tilted current sheet and the limitation of our 2-D simulations in lines 157-166.

A few more minor points:

1. The “Newton Challenge” used a standard Harris sheet current sheet set-up.

Reply: Thanks! The Reviewer is correct that the “Newton Challenge” mostly used a standard Harris current sheet, even though Pritchett (2005) considered the extension to the Lembège-Pellat current sheet in the kinetic aspects of the Newton Challenge. To avoid any misunderstandings, we have changed “The initial condition is the same as in the Newton Challenge” to “The initial condition is the Lembège-Pellat current sheet” (line 240).

2. In the last paragraph of the discussion ‘about several inertial lengths’ should be ‘about seven inertial lengths’.

Reply: Thanks! We have now changed “several” to “seven” because the latter one is more precise (line 216).

3. I think the authors should estimate whether the normal field B_N is small enough in the MMS data to remove the electron tearing stabilization mechanism.

Reply: We thank the Reviewer for this helpful suggestion! Using the criterion given by Liu et al. (2014), we obtain a threshold 0.171 – 0.214 nT for the normal magnetic field, and the normal magnetic field is $B_n = 0.148$ nT at the center of the current sheet – smaller than the threshold.

This shows that the normal magnetic field is small enough so that the electron tearing mode instability is not stabilized by it. A detailed discussion on this issue has now been added into the revised manuscript, see lines 182-198.

Overall, I think the paper is of interest and could be suitable for publication in this journal. The subject problem is of high interest for magnetic reconnection and the wider space physics community. It is particularly challenging to address this problem, but the authors present what is probably the most compelling evidence yet for the driven electron tearing scenario. However, I would like the authors to comment on the differences between the MMS data and their simulations, rather than ignore the limitations. The paper is well written, and contains appropriate references.

References

- Liu, Y.-H., Birn, J., Daughton, W., Hesse, M., & Schindler, K. Onset of reconnection in the near magnetotail: PIC simulations. *J. Geophys. Res. Space Physics* **119**, 9773-9789 (2014).
- Pritchett, P. L. The “Newton Challenge”: Kinetic aspects of forced magnetic reconnection. *J. Geophys. Res. Space Physics* **110**, A10213 (2005).
- Sitnov, M. I., Buzulukova, N., Swisdak, M., Merkin, V. G., & Moore T. E. Spontaneous formation of dipolarization fronts and reconnection onset in the magnetotail. *Geophys. Res. Lett.* **40**, 22-27 (2013).

REVIEWERS' COMMENTS:

Reviewer #1 (Remarks to the Author):

The authors have addressed my questions. The flapping motion of the current sheet indeed adds an additional layer of complication in this event, and I agree with another reviewer that this difference shall be discussed. The readers shall be made aware of this issue. This work paves the path for studying tail-reconnection onset using MMS observation. I recognize its significance and will suggest this version for publication in Nature Communications.

Reviewer #2 (Remarks to the Author):

The authors have thoroughly addressed the concerns raised in my initial review. I believe the paper is suitable for publication.